# Simple and Optimal Greedy Online Contention Resolution Schemes

**Vasilis Livanos**
Department of Computer Science
University of Illinois Urbana-Champaign
Urbana, IL 61801
`livanos3@illinois.edu`

## Abstract

Real-world problems such as ad allocation and matching have been extensively studied under the lens of combinatorial optimization. In several applications, uncertainty in the input appears naturally and this has led to the study of online stochastic optimization models for such problems. For the offline case, these constrained combinatorial optimization problems have been extensively studied, and Contention Resolution Schemes (CRSs), introduced by Chekuri, Vondrák, and Zenklusen, have emerged in recent years as a general framework to obtaining a solution. The idea behind a CRS is to first obtain a fractional solution to a (continuous) relaxation of the objective and then round the fractional solution to an integral one. When the order of rounding is controlled by an adversary, Online Contention Resolution Schemes (OCRSs) can be used instead, and have been successfully applied in settings such as prophet inequalities and stochastic probing.

In this work, we focus on *greedy* OCRSs, which provide guarantees against the strongest possible adversary, an *almighty* adversary. Intuitively, a greedy OCRS has to make all its decisions before the online process starts. We present simple $1/e$ - selectable greedy OCRSs for the single-item setting, partition matroids and transversal matroids, which improve upon the previous state-of-the-art greedy OCRSs of [FSZ16] for these constraints. We also show that our greedy OCRSs are optimal, even for the simple single-item case.

## 1 Introduction

In recent years, problems in Bayesian and stochastic online optimization have attracted significant interest, especially in the field of machine learning. In this setting, we are usually asked to make decisions in an online manner, based on the information available to us so far, and our objective is to minimize our "regret", which is captured by a loss function and describes how much better we could have done if we had all the information available a priori.

In several applications, in which our decision relies also on hidden information, such as deciding whether a user will click an ad they are presented with [AAD+20, XSC+19, BSSX20, BGMS21] or whether a kidney donor is a good match for another patient [BFT13, DS15], our online decision problem naturally becomes stochastic. The inherent uncertainty of such applications makes the study of stochastic optimization all the more significant, and demonstrating algorithms with reasonable competitive ratios when one has knowledge of the underlying distributions illustrates the importance of learning distributions in AI.

One can consider a simpler setting in which our decision at each time step is a binary choice. In one such formulation, elements arrive in an online manner and we have to select a subset of them, subject to certain combinatorial constraints. The simplest example of such a constraint is when we

36th Conference on Neural Information Processing Systems (NeurIPS 2022).

only want to select a single element. Under this formulation, the elements reveal one after the other in an online manner whether they are "active" or not, and our binary decision is whether to select an active element, with the constraint of being able to select at most one active element. In this paper, we study this problem and its generalizations. Famous examples of such settings are prophet inequality problems [KS77, KW12, EFGT20] and secretary problems [Dyn63], which have found application in the design of posted-price mechanisms and auctions, among others.

One approach that has seen plenty of success for these problems is to use the known distributional information to obtain a continuous relaxation of the objective. One can then solve this relaxation and get an optimal fractional solution $x^*$, which corresponds to the marginals of the elements under the optimal distribution. Thus, the optimal value of the relaxation constitutes an upper bound to the performance of any online (or even offline) algorithm. Afterwards, $x^*$ is used to devise an online algorithm in order to maximize the value of the subset of elements selected. It is easy to see that this algorithm essentially corresponds to an online rounding procedure for $x^*$.

In this paper, we contribute provably optimal algorithms for the problem of rounding an optimal solution to a linear program in an online manner, for a range of fundamental constraints. The online nature of our rounding is well-motivated in the field of AI due to the fact that we are not able to control the arrival order of the agents. Our algorithms apply for the single-item setting, for partition matroids, as well as transversal matroids which have been used extensively to model matching markets [DRS09, CFMP, BIK07].

## 1.1 Online Contention Resolution Schemes

Such rounding algorithms have recently been used to obtain several optimal and interesting results [CVZ11, FSZ16, AW18, EFGT20, BZ20, RS17, CL21], and have more applications in online mechanism design and posted pricing mechanisms [CHMS10, HKS07]. General rounding algorithms for offline problems are called *Contention Resolution Schemes (CRSs)* and were introduced by Chekuri, Vondrák and Zenklusen [CVZ11] with the purpose of maximizing a submodular function. A CRS is defined with respect to a constraint family. Examples of such combinatorial constraints include selecting an independent set in a given matroid, selecting a feasible matching in a given a graph in which the elements correspond to edges, or selecting a feasible set of elements subject to a knapsack constraint, where each element is associated with a size. Chekuri, Vondrák and Zenklusen [CVZ11] gave the first CRSs for all aforementioned constraint families as well as other constraints. For a given fractional point $x^*$, the main idea behind CRSs is to first obtain a random set $R$, drawn from the product distribution with marginals $x^*$, hence called the *active elements*. Since $R$ may be infeasible with respect to the constraints, the CRS proceeds to "drop" specific elements from $R$ and obtain a new, feasible, set $R' \subseteq R$.

While the general applicability of the CRS approach is remarkable, they are unfortunately not useful for Bayesian and stochastic online optimization problems. In particular, one can utilize CRSs when they have the ability to choose the order in which they obtain information about the underlying ground set of elements, as CRSs round a fractional point $x^*$ in a particular order to obtain a feasible solution.

To overcome the inherently offline nature of CRSs, Feldman, Svensson and Zenklusen [FSZ16] introduced the notion of *Online Contention Resolution Schemes (OCRSs)*, applicable in a variety of online settings in Bayesian and stochastic online optimization, such as prophet inequalities [LS18, RS17, CL21], stochastic probing [ASW16, GN13, GNS16, GNS17], and posted pricing mechanisms [HKS07]. Surprisingly, OCRSs yield constant-factor competitive ratios for several interesting feasibility constraints.

All of the results presented in [FSZ16] are based on a special subclass of OCRSs called *greedy OCRSs*. Intuitively, a greedy OCRS fixes a downward-closed subfamily of feasible sets $\mathcal{F}$ before the online process starts. During the online process, the greedy OCRS maintains a subset $S$ of the elements which is feasible in $\mathcal{F}$, and then greedily accepts any active element $i$ if $S \cup \{i\}$ is also feasible in $\mathcal{F}$, i.e. if $i$ does not violate feasibility, with respect to $\mathcal{F}$, of the set maintained by the greedy OCRS. One can easily see that the final set at the end of the online process is feasible by construction.

Even though greedy OCRSs offer suboptimal performance guarantees with respect to (non-greedy) OCRSs, as we will see, their study remains interesting for two important reasons. First, greedy OCRSs are inherently simpler than their non-greedy counterparts. Usually, to obtain an optimal

non-greedy OCRS for a non-trivial constraint, one has to use linear programming duality, as in the approach of [LS18]. This leads to a non-intuitive algorithm which, in many situations, can be difficult to implement[1]. In short, greedy OCRSs are simpler to implement and more intuitive.

Furthermore, greedy OCRSs provide guarantees against an *almighty adversary* who has knowledge of the future as well as any random coins used by the algorithm. This property is crucial for applications that require the algorithm to compare against an almighty adversary. One such example is [BDP22] in which the authors study the "delegation gap" of the generalized Pandora's box problem and in fact reduce the problem to the design of an OCRS which is necessarily greedy. To the best of our knowledge, this result is the first example of an application in which non-greedy OCRSs cannot be applied and a greedy OCRS is needed.

## 1.2 Our contributions

In this paper, we analyze the performance of greedy OCRSs and provide the first provably optimal greedy OCRS for the single-item setting, partition matroids and transversal matroids.

We have four main contributions:

- We design a $1/e$-selectable greedy OCRS for the single-item setting (Theorem 1.1).
- We show that our greedy OCRS extends naturally to partition matroids[2] (Corollary 1.2).
- We proceed to show that no greedy OCRS can be $(1/e + \varepsilon)$-selectable, for any $\varepsilon > 0$, even for the single item setting. This, combined with our first contribution, shows that our $1/e$-selectable greedy OCRS is the best possible (Theorem 1.3).
- We extend our greedy OCRS to transversal matroids[3] as well, and show that the selectability can be increased to $1 - 1/e$ for special cases of transversal matroids (Theorem 1.4).

Our results improve upon the $1/4$-selectable OCRSs of [FSZ16] for all the constraints discussed here.

As a corollary, our work presents the first instance of a dichotomy between the best possible guarantees by greedy OCRSs and (non-greedy) OCRSs, since a $1/2$ (non-greedy) OCRS is known for the single-item setting [Ala14].

We proceed with our four main results. The proof of the following theorem is found in Section 3.

**Theorem 1.1.** *There exists a $1/e$-selectable (randomized) greedy OCRS for the single-item setting.*

Next, we extend the single-item greedy OCRS to a partition matroid constraint, by decomposing the partition matroid into single-item instances, running the greedy OCRS above and accepting an active element if and only if it is independent in the corresponding single-item instance of the decomposition.

**Corollary 1.2.** *There exists a $1/e$-selectable (randomized) greedy OCRS for partition matroids.*

We complement the results above by also showing that it is tight. The proof of the following theorem can be found in Section 4.

**Theorem 1.3.** *For every $\varepsilon > 0$, there exists no greedy OCRS for the single-item setting that selects an active element $i$ with probability at least $1/e + \varepsilon$ for all $i \in \mathcal{N}$.*

Finally, we extend Theorem 1.1 to a more general class of matroids, transversal matroids, and strengthen it for the special case in which every element's neighborhood has size at least 3. The proof of the following theorem is found in Section 5.

**Theorem 1.4.** *Let $\mathcal{M} = (U, \mathcal{I})$ be a transversal matroid represented by a bipartite graph $G = (U \cup V, E)$. Then, there exists a $1/e$-selectable (randomized) greedy OCRS $\pi$ for $\mathcal{M}$. Furthermore, if for every element $u \in U$ we have $|N(u)| \geq 3$, where $N(u)$ is the set of neighbours of $u$ in $G$, then $\pi$ is a $(1 - 1/e)$-selectable (randomized) greedy OCRS for $\mathcal{M}$.*

---

[1]For example if the OCRS is used for an application in which one has to account for the strategic behaviour of the agents.

[2]A partition matroid consists of a partition of the elements into disjoint sets $A_1, \ldots, A_k$ such that a subset of the elements $S$ is independent if and only if $|S \cap A_j| \leq 1$ for every $1 \leq j \leq k$.

[3]A transversal matroid consists of a bipartite graph $G = (A \cup B, E)$, in which the set of elements is $A$ and a set $S \subseteq A$ is independent if and only if there exists a matching in $G$ that covers $S$.

## 1.3 Related work

Since their introduction [CVZ11], Contention Resolution Schemes (CRSs) have found several applications. Applications of CRSs in Bayesian mechanism design and posted price mechanisms [CHMS10] can be found in [CVZ11]. Later, Yan [Yan11] connected mechanism design with the notion of correlation gap [ADSY12]. OCRSs were developed [FSZ16] with applications to Bayesian mechanism design as one of the main motivations as they directly translate to competitive ratios for the prophet inequality problem [FSZ16, Rub16, RS17, CL21]. In fact, Alaei's work on uniform matroids [Ala14] precedes [FSZ16] and can be seen as an OCRS, even though it is formulated differently. Random order CRSs (ROCRSs) were introduced in [AW18] and yield improved bounds when the arrival order is random.

As stated previously, Feldman, Svensson and Zenklusen [FSZ16] gave the first greedy OCRS for matroids, which is $1/4$-selectable. Lee and Singla [LS18] showed a reverse connection between OCRSs and prophet inequalities, obtaining a $1/2$-selectable (non-greedy) OCRS for matroids and a $(1 - 1/e)$-selectable ROCRS for the single item setting. Adamczyk and Wlodarczyk [AW18] obtained several results, including a $1/k+1$-selectable ROCRS for the intersection of $k$ matroids. For matchings, Ezra et al [EFGT20] designed a $0.337$-selectable OCRS for bipartite graphs, while Bruggmann and Zenklusen [BZ20] developed optimal monotone CRSs via a novel polyhedral approach.

This work is connected to stochastic optimization, online algorithms, mechanism design and submodular optimization, all of which have extensive literature. There have been several surveys on the topic [Gup17, Luc17, CFH+19, Din13, HK92], as well as a survey on random-order models in general [GS20]. Singla's thesis [Sin18] has connections to several of the topics discussed here. On the application side, prophet inequality and secretary problems have received significant attention in the last years, due to their connections with Bayesian mechanism design and posted price mechanisms [GM66, HK82, Ker86, Dyn63, KS77, KW12, ACK17, CFH+17, CSZ20, AEE+17, EHLM15, HKS07, EHKS18, FZ18], while ROCRSs have found several applications to *stochastic probing* [ASW16, GN13, GNS16, GNS17, BSZ19, AN16]. Recently, Dughmi [Dug20, Dug21] showed the equivalence between the existence of constant-factor *universal* OCRSs and a constant-factor approximation to the famous matroid secretary problem [BIKK18]. Apart from OCRSs, the other main technique that has emerged for proving prophet inequalities and guarantees for posted-price mechanisms is the "balanced prices" framework [KW12, FGL15, DFKL20, DKL20]. We refer the reader to a survey by Lucier [Luc17] for more information on this separate technique.

Independently, [FLT+21] study the problem of designing an *oblivious* OCRS [4] for the same setting and obtain a similar result, showing that there exists a $1/e$-selectable oblivious OCRS and no oblivious OCRS can be $(1/e + \varepsilon)$-selectable for any $\varepsilon > 0$. We note that the two results (and schemes) are very different. In fact, their OCRS is not greedy, while ours is not oblivious. Whether one can achieve similar guarantees with greedy and oblivious OCRSs for more general settings is an interesting open problem.

## 1.4 Roadmap

We begin in Section 2 with some background. Then, in Section 3, we present our first main result, the $1/e$-selectable greedy OCRS for the single-item setting and partition matroids. Then, in Section 4, we show that our greedy OCRS is optimal. We proceed with our greedy OCRS for transversal matroids in Section 5, which also achieves the optimal $1/e$ selectability and show it performs even better under mild assumptions on the structure of the transversal matroid. Finally, we present our experimental results in Section 6, demonstrating how our schemes outperform the best previously known schemes for randomly generated as well as specific worst-case instances. All omitted proofs can be found in the Appendix which, along the code used for the experiments, is provided in the supplementary material.

---

[4]An CRS (or OCRS) is called oblivious if and only if it does not make use of the fractional point $x$, i.e. if for every $S \subseteq \mathcal{N}$, the distribution of $\pi_x(S)$ and the distribution of $\pi_y(S)$ are identical for any two fractional points $x, y \in \mathcal{P}_{\mathcal{F}}$.

## 2 Preliminaries

Before we proceed, we present the formal definitions of CRSs, OCRSs and greedy OCRSs and briefly describe a $1/4$-selectable single item OCRS by [FSZ16].

Let $\mathcal{N}$ be a finite ground set. A constraint family over $\mathcal{N}$ is a subset $\mathcal{I} \subseteq 2^{\mathcal{N}}$; a set $S \in \mathcal{I}$ is called feasible, while a set $S \notin \mathcal{I}$ is called infeasible. We say $\mathcal{P}_{\mathcal{I}} \subseteq [0,1]^{\mathcal{N}}$ is a polyhedral relaxation of $(\mathcal{N}, \mathcal{I})$ if $\mathcal{P}_{\mathcal{I}}$ is a polyhedron and $\mathbb{1}_S \in \mathcal{P}_{\mathcal{I}}$ for all $S \in \mathcal{I}$ (here $\mathbb{1}_S$ is the characteristic vector of $S$).

Given a polyhedral relaxation $\mathcal{P}_{\mathcal{I}}$ of a constraint $(\mathcal{N}, \mathcal{I})$ and a point $x \in \mathcal{P}_{\mathcal{I}}$, a natural question is whether we can round $x$ in order to obtain a feasible set $S \in \mathcal{I}$. One way to achieve this is via Contention Resolution Schemes, which we define below.

**Definition 2.1** (Contention Resolution Scheme [CVZ11]). Let $b, c \in [0,1]$. A $(b,c)$-balanced *Contention Resolution Scheme* $\pi$ for $\mathcal{P}_{\mathcal{I}}$ is a procedure that for every $x \in b \cdot \mathcal{P}_{\mathcal{I}}$ and $A \subseteq \mathcal{N}$, returns a random set $\pi_x(A) \subseteq A \cap \text{support}(x)$ and satisfies the following properties:

1. $\pi_x(A) \in \mathcal{I}$ with probability 1, $\quad \forall A \subseteq \mathcal{N}, x \in b \cdot \mathcal{P}_{\mathcal{I}}$, and

2. for all $i \in \text{support}(x)$, $\Pr[i \in \pi_x(R(x)) \mid i \in R(x)] \geq c$, $\quad \forall x \in b \cdot \mathcal{P}_{\mathcal{I}}$,

where $R(x) \subseteq \mathcal{N}$ denotes a random set in which every element $i \in \mathcal{N}$ appears independently with probability $x_i$.

The scheme is said to be *monotone* if $\Pr[i \in \pi_x(A_1)] \geq \Pr[i \in \pi_x(A_2)]$ whenever $i \in A_1 \subseteq A_2$.

For the remainder of this paper, we drop the subscript in $\mathcal{P}_{\mathcal{I}}$ and simply write $\mathcal{P}$ whenever the constraint is clear from context.

CRSs are offline rounding schemes. In the case where the arrival order of the elements is selected by an adversary, we can use the following notion of *Online Contention Resolution Schemes (OCRS)* to round $x$.

**Definition 2.2** (Online Contention Resolution Scheme (OCRS) [FSZ16]). For an online selection setting where a point $x \in \mathcal{P}$ is given, we draw a random subset of the elements $R(x)$, in which each element $i$ appears independently with probability $x_i$. We call $R(x)$ the set of *active* elements. Afterwards, we observe whether the element $e \in \mathcal{N}$ are active ($e \in R(x)$), one by one, and have to immediately and irrevocably decide whether to select an element or not before the next element is revealed. An *Online Contention Resolution Scheme* for $\mathcal{P}$ is an online algorithm which selects a subset $I \subseteq R(x)$ such that $\mathbb{1}_I \in \mathcal{P}$.

A scheme is called a *Random Order Contention Resolution Schemes (ROCRS)* if, instead of being chosen by an adversary, the arrival order of the elements is chosen uniformly at random. Adamczyk and Wlodarczyk present several interesting results on ROCRSs in [AW18]. In the case of adversarial arrival order, however, one can distinguish between three different adversaries in terms of the information they have at their disposal. An *offline adversary*, which is the weakest of the three, has to fix an ordering of the elements before any of the elements are revealed. An *almighty adversary*, the most powerful one, has access to the realizations of all random events; both the set of active elements and any potential random bits the algorithm may use. Therefore, an almighty adversary can predict the algorithm's behaviour and choose a truly worst-case ordering of the elements for the particular algorithm. In between the two extremes is the *online adversary*. An online adversary's choices can only depend on the realizations of the elements that have appeared so far. In other words, the adversary has, at any step, exactly the same information as the algorithm, and their decision as to which element to reveal at step $i$ can only depend on the realizations of the elements revealed in steps 1 through $i-1$.

We also define the notion of a *greedy* OCRS, which provide guarantees with respect to an almighty adversary.

**Definition 2.3** (Greedy OCRS [FSZ16]). Let $\mathcal{P} \subseteq [0,1]^n$ be a relaxation of the feasible sets $\mathcal{F} \subseteq 2^{\mathcal{N}}$. An OCRS $\pi$ for $\mathcal{P}$ is called a *greedy OCRS* if, for any $x \in \mathcal{P}$, $\pi$ defines a down-closed subfamily of feasible sets $\mathcal{F}_{\pi,x} \subseteq \mathcal{F}$, and it selects an active element $e$ when it arrives if, together with the set of elements already selected, the resulting set is in $\mathcal{F}_{\pi,x}$. We say that $\pi$ is a randomized greedy OCRS if, given $x$, the choice of $\mathcal{F}_{\pi,x}$ is randomized. Otherwise, we say that $\pi$ is a deterministic greedy OCRS.

For the remainder of this paper, we drop the subscript in $\mathcal{F}_{\pi,x}$ and simply write $\mathcal{F}_x$ or $\mathcal{F}$, whenever $\pi$ and $x$ are clear from context.

Intuitively, we say a greedy OCRS is $c$-selectable if and only if an active element $e \in R(x)$ can be included in the currently selected elements $I \subseteq R(x)$ and maintain feasibility with probability at least $c$.

**Definition 2.4** ($c$-selectability)**.** Let $c \in [0,1]$. A greedy OCRS for $\mathcal{P}$ is $c$-selectable if and only if for any $x \in P$ we have

$$\Pr\left[I \cup \{e\} \in \mathcal{F}_x \forall I \subseteq R(x), I \in \mathcal{F}_x\right] \geq c \qquad \forall e \in \mathcal{N}.$$

Notice that a $c$-selectable greedy OCRS guarantees that each active element $e$ is selected with probability at least $c$, even against the almighty adversary. We should note that the randomness in the above definition is with respect to both the randomness of $R(x)$ and also any potential randomness the greedy OCRS might use to decide upon $\mathcal{F}_x$.

Next, we briefly describe the $1/4$-selectable single item greedy OCRS by [FSZ16]. Given a fractional point $x$ such that $\sum_{i=1}^n x_i \leq 1$, the greedy OCRS will, at step $i$, observe whether element $i$ is active or not. If it is active, the greedy OCRS will choose to select with probability $1/2$ or discard it and move on to the next element. Since each element is active with probability $x_i$ and is selected with probability $x_i/2$, the expected number of selected elements is at most half, and thus, by Markov's inequality, the probability the greedy OCRS selects no elements is at least $1/2$. Therefore, for every element $i$, we reach $i$ without having selected an element with probability at least $1/2$ and we select $i$, given that it is active, with probability $1/2$, for an overall selectability of $1/4$.

We should note that for the single item setting there exists a $1/2$-selectable OCRS [Ala14] but, crucially, it is not greedy. In fact, we show in the Section 4 that there is no $1/2$-selectable greedy OCRS for the single item setting.

## 3   An $1/e$-selectable greedy OCRS for the single-item setting

This section is dedicated to proving Theorem 1.1. Before we begin, we need the following lemma.

**Lemma 3.1.** *Let $a_1, \ldots, a_k \in [0,1]$. Then*

$$\ln\left(1 - \frac{a_k}{2}\right) + \sum_{j=1}^{k-1} \ln\left(1 - a_j + \frac{a_j^2}{2}\right) \geq -a_k - \sum_{j=1}^{k-1} a_j$$

Next, consider a ground set $\mathcal{N} = \{e_1, e_2, \ldots, e_n\}$, and let $\mathcal{M} = (\mathcal{N}, \mathcal{I})$ be the uniform matroid of rank 1 with respect to $\mathcal{N}$, i.e. $\mathcal{I} = \{\{e_i\} \mid e_i \in \mathcal{N}\}$. Let $\mathcal{P}$ be the following polyhedral relaxation of $\mathcal{M}$:

$$\mathcal{P} = \left\{ x \in [0,1]^n \;\middle|\; \sum_{i=1}^n x_i \leq 1 \right\}$$

For a given $x \in \mathcal{P}$, let $\pi = \pi_x$ denote the OCRS we will create. $\pi$ will draw a random set $R(q)$ where each element $e_i$ appears in $R(q)$ independently with some probability $q_i$. The family of feasible subsets is

$$\mathcal{F}_{\pi,x} = \{\{e_i\} \mid e_i \in R(q)\}.$$

We set $q_i = 1 - x_i/2$ for all $e_i \in \mathcal{N}$. Afterwards, $\pi$ selects the first element $e_i$ that is active and that $\{e_i\} \in \mathcal{F}$.

**Lemma 3.2.** $\pi$ *is a randomized greedy OCRS.*

Next, we quantify the probability that each element is selected by $\pi$, given that it is active.

**Lemma 3.3.** $\pi$ *selects every element $e_i \in \mathcal{N}$, given that it is active, with probability at least $1/e$.*

*Proof.* We relabel the elements of $\mathcal{N}$ so that each $e_i$ arrives in the $i$-th step. Consider an element $e_i \in \mathcal{N}$. Given that $e_i$ is active, since $\pi$ is a greedy OCRS, $\pi$ will select $e_i$ if and only if it has not selected any elements before $e_i$ and also $\{e_i\} \in \mathcal{F}_{\pi,x}$. Recall that we have $\{e_i\} \in \mathcal{F}_{\pi,x}$ with

probability exactly $q_i = 1 - x_i/2$. Furthermore, for every element $e_j$ where $j < i$, it needs to be the case that we avoid having both $\{e_j\} \in \mathcal{F}_{\pi,x}$ and also $e_j$ coming up active. This happens with probability $1 - x_j \cdot (1 - x_j/2) = 1 - x_j + x_j^2/2$ for every $e_j$ where $j < i$. Overall, if we denote by $r_i$ the probability that $e_i$ is selected by $\pi$, given that it is active, we have

$$\ln r_i = \ln\left(\left(1 - \frac{x_i}{2}\right) \cdot \prod_{j=1}^{i-1}\left(1 - x_j + \frac{x_j^2}{2}\right)\right) = \ln\left(1 - \frac{x_i}{2}\right) + \sum_{j=1}^{i-1}\ln\left(1 - x_j + \frac{x_j^2}{2}\right)$$

$$\geq -x_i - \sum_{j=1}^{i-1} x_j \geq -1,$$

where the first inequality follows from Lemma 3.1 and the second inequality follows from $\sum_i x_i \leq 1$. Therefore $r_i \geq 1/e$, for all $i \in \mathcal{N}$. □

From Lemmas 3.2 and 3.3, it follows that $\pi$ is a $1/e$-selectable (randomized) greedy OCRS for $\mathcal{P}$.

**Remark 3.4.** In a personal communication, Jan Vondrák devised an alternate scheme for the problem, after we notified him of our scheme. With his consent [Von], we have included this alternate scheme in the Appendix, which can be found in the supplementary material.

## 4 $1/e$ **is tight**

In this section, we present the proof of Theorem 1.3. Consider the instance where $x_i = 1/n$ for all $e_i \in \mathcal{N}$, where $n = |\mathcal{N}|$, and let $A$ denote the set of active elements. Any greedy OCRS $\pi$ will select a subset $S$ of $\mathcal{F} = \{e_i \mid e_i \in \mathcal{N}\}$ with some probability $\alpha_S$, and then accept the first element in $S$ that comes up active. What is the worst-case probability that an element from $S$ will be selected? This is minimized for the element in $S$ which is last in the arrival order, which has a probability of being selected exactly equal to $(1 - 1/n)^{|S|-1}$, because the OCRS is greedy, and it would select an element from $S$ which arrived earlier, if it came up active. Therefore, no greedy OCRS can guarantee, for any $S \subseteq \mathcal{F}$, that an element $e \in \mathcal{N}$ will be selected, when $e \in A$, with probability greater than $(1 - 1/n)^{|S|-1}$. Thus, for any $e \in \mathcal{N}$ and any greedy OCRS $\pi$, we have

$$\Pr\left[e \in \pi(A) \mid e \in A\right] \leq \sum_{\substack{S \subseteq \mathcal{N} \\ e \in S}} \alpha_S \left(1 - \frac{1}{n}\right)^{|S|-1}$$

$$= \sum_{k=1}^n \left(1 - \frac{1}{n}\right)^{k-1} \sum_{\substack{S \subseteq \mathcal{N} \,:\, |S|=k \\ e \in S}} \alpha_S. \tag{1}$$

Next, for a greedy OCRS $\pi$ to be $c$-selectable, it needs to guarantee that $\min_{e \in \mathcal{N}} \Pr\left[e \in \pi(A) \mid e \in A\right] \geq c$. Therefore, if we show that

$$\min_{e \in \mathcal{N}}\left\{\sum_{k=1}^n \left(1 - \frac{1}{n}\right)^{k-1} \sum_{\substack{S \subseteq \mathcal{N} \,:\, |S|=k \\ e \in S}} \alpha_S\right\} \leq c,$$

by (1) it follows that $\pi$ cannot be $(c + \varepsilon)$-selectable for any $\varepsilon > 0$.

**Lemma 4.1.**

$$\min_{e \in \mathcal{N}}\left\{\sum_{k=1}^n \left(1 - \frac{1}{n}\right)^{k-1} \sum_{\substack{S \subseteq \mathcal{N} \,:\, |S|=k \\ e \in S}} \alpha_S\right\} \leq \left(1 - \frac{1}{n}\right)^{n-1}.$$

By Lemma 4.1, since $\lim_{n \to \infty} (1 - 1/n)^{n-1} = 1/e$, it follows that there exists no greedy OCRS for $\mathcal{P}$ that selects an element $e$, when active, with probability at least $1/e + \varepsilon$ for all $e \in \mathcal{N}$.

# 5 Extension to Transversal Matroids

In this section, we prove Theorem 1.4. Let $\mathcal{M} = (U, \mathcal{I})$ be a transversal matroid and $G = (U \cup V, E)$ denote the underlying bipartite graph, where $|U| = n$. We know that a subset $S \subseteq U$ is independent if and only if there exists a matching in $G$ that covers $S$. Let $\mathcal{P}$ be the natural polyhedral relaxation of $\mathcal{M}$. For a given $x \in \mathcal{P}$, let $\pi = \pi_x$ be the greedy OCRS we will create. For each $v \in V$, $\pi$ will draw a random set $R_v \subseteq N(v)$, in which each element $u \in U$ appears with probability $q_u$. For every $u \in U$, let $N(u)$ denote the set of neighbors of $u$ in $G$. Then, we set

$$q_u = 1 - \left(1 - \frac{1 - e^{-x_u}}{x_u}\right)^{\frac{1}{|N(u)|}}.$$

It is easy to see that $q_u \in [0, 1]$ for every $|N(u)| \geq 1$, and thus $q_u$ is well-defined.

Next, we create a down-closed subfamily of feasible sets by taking all possible combinations of sets created by taking at most one element from each $R_v$ and then taking the union of all such elements. Specifically,

$$\mathcal{F}_{\pi,x} = \left\{ S = \{u_1, \ldots, u_k\} \subseteq U \mid \exists\, T = \{v_1, \ldots, v_k\} \subseteq V \text{ s.t. } u_j \in R_{v_j}, \ \forall j \in \{1, \ldots, k\} \right\}.$$

Any set $S$ in $\mathcal{F}$ is clearly an independent set of $\mathcal{M}$, as the constraints guarantee that there always exists a matching in $G$ that covers $S$. During the online process, $\pi$ starts with a set of selected elements $S = \emptyset$, and greedily selects an active element $u$ if $S + u \in \mathcal{F}$.

The proof of the following lemma is identical to the proof of Lemma 3.2 and follows from the discussion above.

**Lemma 5.1.** $\pi$ is a randomized greedy OCRS.

Next, we again lower bound the selection probability of an active element.

**Lemma 5.2.** $\pi$ selects every element $u \in U$, given that it is active, with probability at least $1/e$. Furthermore, if $|N(u)| \geq 3$ for all $u \in U$, $\pi$ selects every element $u \in U$, given that it is active, with probability at least $1 - 1/e$.

We conclude that $\pi$ is a $1/e$-selectable greedy OCRS for $\mathcal{M}$ and that if $|N(u)| \geq 3$ for every $u \in U$, $\pi$ is a $(1 - 1/e)$-selectable greedy OCRS for $\mathcal{M}$.

# 6 Experimental Results

In this section, we present our experimental results which give a quantitative view into how our greedy OCRSs outperform the former state-of-the-art greedy OCRSs by [FSZ16]. For the constraint system, we consider both the single-item setting as well as randomly-generated transversal matroid constraints. The code used to run these experiments can be found in the supplementary material.

For our experiments, we have the following parameters:

- N: The number of elements. For the single item setting, this varies from 10 to 100. For the transversal matroid constraint, this is set to 50.

- ITERATIONS: For a fixed $x$, the number of times we simulate the realization of the elements. This is used to approximate the selectability of each element within a reasonable estimate and is set to $200,000$.

- REPETITIONS: The number of times we run the entire experiment from scratch. This has no effect on the uniform instance of the single-item setting or the transversal matroid constraint, but affects the random vector $x$ in the randomly-generated instance of the single-item setting, and is set to 10.

We focus on two metrics: the *minimum selectability* and the *average selectability* of each element. For a fixed $x$, the minimum (resp. average) selectability records the minimum (resp. average) over all elements of the fraction of the times an element was selected by our greedy OCRS over the number of times it was active. While the average gives an idea of what selectability to expect for most elements, the minimum selectability provides a worst-case estimate.

For both types of constraints, we can see that our greedy OCRSs outperform the $1/4$-selectable single-item greedy OCRS by [FSZ16] in the minimum selectability metric and performs substantially better in the average selectability metric.

## 6.1 Single-Item Setting

We consider two different types of instances for the single-item setting: the *uniform instance*, where $x_i = 1/n$ for every element $i \in N$, as well as a *randomly-generated instance*, in which $x$ is a random vector of length $n$ with the constraint that $\sum_i x_i = 1$.

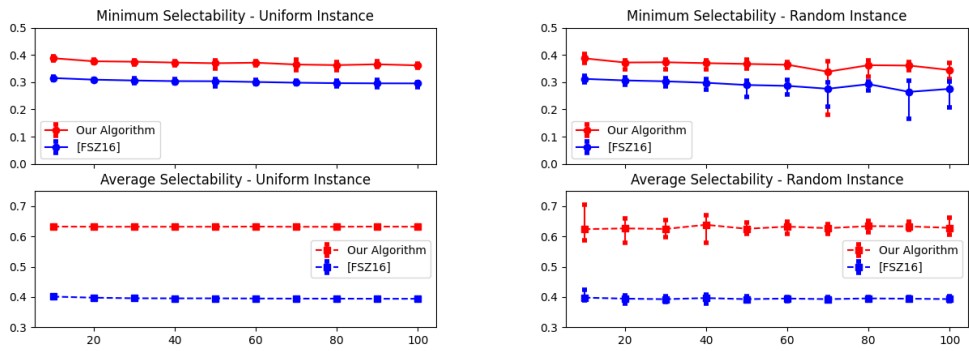

Figure 1: N = 5 to 100, ITERATIONS = 200,000, REPETITIONS = 10

## 6.2 Transversal Matroid

For the transversal matroid constraint, we consider three different types of instances: a transversal matroid with $|V| = 2$, $|N(u)| = 2$ and $x_u = 2/n$ for all vertices $u \in U$, a transversal matroid with $|V| = 3$, $|N(u)| = 3$ and $x_u = 3/n$ for all vertices $u \in U$, as well as a transversal matroid with $|V| = 4$, $|N(u)| = 4$ and $x_u = 4/n$ for all vertices $u \in U$.

While there are many options for the choice of what transversal matroid to pick, we selected these "uniform" instances of transversal matroids to illustrate how the selectability changes as $|N(u)|$ increases. Since a transversal matroid with $|V| = 1$, $|N(u)| = 1$ and $x_u = 1/n$ for all vertices $u \in U$ is equivalent to a uniform instance of the single-item setting, we do not consider it explicitly, but we plot it together with the previous three instances. The x-axis in the figures below corresponds to the size of $|N(u)|$.

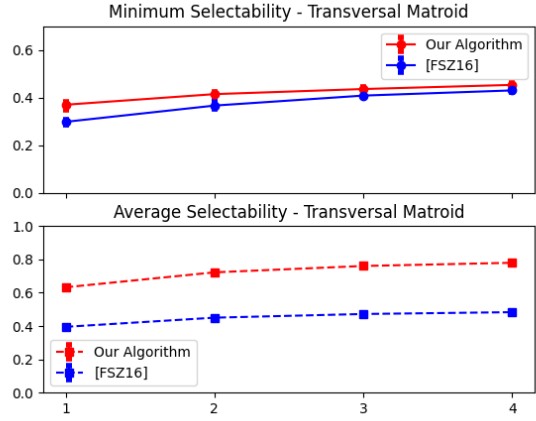

Figure 2: N = 50, ITERATIONS = 200,000

## Acknowledgments and Disclosure of Funding

The author would like to thank Chandra Chekuri, Ruta Mehta and Jan Vondrák for guidance and helpful discussions.

Work on this paper partially supported by NSF grant CCF-1910149.

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
