# OpenReview forum: "Simple and Optimal Greedy Online Contention Resolution Schemes"
_NeurIPS.cc/2022/Conference — NeurIPS 2022 Accept_

### Official Review · Reviewer_KzEE · 2022-07-11

**Rating:** 7
**Confidence:** 3
**Soundness:** 4 excellent
**Presentation:** 3 good
**Contribution:** 3 good

**Summary:**

The paper studies the problem of designing greedy online content resolution schemes (OCRSs). The author provide an OCRS that provides a $\frac{1}{e}$-selectable greedy strategy for single-item settings and partition matroids. Then, they provide a similar result for transversal matroids. In this case, they also provide a better result for specific settings.
Moreover, they show that these results are tight for greedy OCRC. Finally, they provides a experimental analysis with a comparison with previous works.



**Questions:**

Why is it an important property for OCRSs being greedy? Is it simply because they work against an almighty adversary?

**Limitations:**

Yes.

**Strengths And Weaknesses:**

strengths: The paper provides better results than previous work. Moreover, the authors show that this bound is tight. The paper is easy to follow and provides a detailed analysis of previous works.

Weakness: The derivation of the technical results do not seem too involved.

---

> ### Author Response · Authors · 2022-08-02
> **Rebuttal Response**
>
> The reviewer has raised some very important questions regarding the benefits of using a greedy vs a non-greedy OCRS and we would like to thank them for their input. In general, greedy OCRSs are inherently much simpler than non-greedy OCRSs. Usually, to obtain a non-greedy OCRS for a non-trivial constraint, one has to go through the dual of an LP which captures the problem, as is the approach of Lee and Singla [LS18]. This leads to a non-intuitive algorithm which, in many situations, can be difficult to implement (i.e. when the OCRS is used for a problem from Algorithmic Game Theory, in which one has to account for the agents' strategic behaviour).
> Greedy OCRSs are simpler and more intuitive.
>
> Furthermore, the known $1/2$-selectable OCRS due to Lee and Singla [LS18] does not provide any guarantees against an almighty adversary who has knowledge of the future realizations and the algorithm's randomness, but rather against an online adversary who at any point in time has the same information as the algorithm. In this context, for the constraints considered, our OCRS provides a stronger guarantee, since it holds against a more powerful adversary.
>
> In addition, we would like to point out an application which crucially requires the OCRSs used to be greedy. Between the submission time and now, we were made aware of a paper titled "Delegated Pandora's Box" by Bechtel, Dughmi and Patel. They study the "delegation gap" of the generalized Pandora's box problem and in fact reduce the problem to the design of an OCRS which is necessarily greedy. In their model, they need a greedy OCRS since the reduction forces the algorithm to play against the almighty adversary. This result is the first example of an application in which non-greedy OCRSs cannot be applied and a greedy OCRS is needed. We have revised the write-up and now cite this paper.

---

### Official Review · Reviewer_CZba · 2022-07-12

**Rating:** 5
**Confidence:** 4
**Soundness:** 4 excellent
**Presentation:** 3 good
**Contribution:** 2 fair

**Summary:**

An Online Contention Resolution Scheme (OCRS) is a rounding technique designed for online optimization problems especially in Bayesian optimization (example prophet inequalities). The paper focuses on Greedy online contention resolution schemes and provides an optimal greedy OCRS for the single-item setting as well as for more general constraint sets such as partition matroids and transversal matroids. The paper also demonstrates a matching lower bound.

**Questions:**

1. In Definition 2.1, property 2,  what is R(x)? Did you mean A there?
2. In Definition 2.1, could you clarify the order of quantifiers? Are properties 1 and 2 simultaneously satisfied for all sets A and all points x?
3. In Definition 2.1, property 2, should the RHS be c*x_i?
4. In Definition 2.2, clarify that the random subset R(x) is drawn so that the marginal probability of i \in R(x) = x_i.
5. In line 246, typo - 1 - x_j(1-x_j / 2)
6. In line 261, “no greedy OCRS cannot” -> “no greedy OCRS can”


**Strengths And Weaknesses:**

The paper provides an elegant greedy OCRS for the single-item setting. The scheme itself is very simple and simply selects in F each element i with probability q_i := 1 - x_i/2. The algorithm then selects the first item (in adversarial order) that is active and selected in F.
Since the work of Feldman etal (that introduced OCRS) also introduced the notion of greedy OCRSs back in 2015, it is rather satisfying to see such a simple scheme that is provably optimal (among greedy schemes).

I believe that while interesting, the paper is of limited interest to the Neurips community and would be better suited at a TCS venue.

---

> ### Author Response · Authors · 2022-08-02
> **Rebuttal Response**
>
> We would like to thank the reviewer for raising several questions regarding the write-up, which we address below.
>
> -In Definition 2.1, property 2, what is R(x)? Did you mean A there?
>
> $x$ is a vector in $[0,1]^n$, and $R(x)$ is a random set from $2^\mathcal{N}$, where $\mathcal{N}$ is the set of elements, in which each element $i$ appears with probability $x_i$. We did mean to write $R(x)$, but should have included this clarification, which we have done now.
>
> -In Definition 2.1, could you clarify the order of quantifiers? Are properties 1 and 2 simultaneously satisfied for all sets A and all points x?
>
> You are correct, properties 1 and 2 are satisfied simultaneously for all sets $A$ and points $x$. One could think of both quantifiers as being in front of each property.
>
> -In Definition 2.1, property 2, should the RHS be c*x_i?
>
> You have the correct intuition, but your formulation is for the overall probability that $i \in \pi_x(R(x))$. Our formulation considers the conditional probability, given  that $i \in R(x)$, which happens with probability $x_i$, and thus the right side of c is correct.
>
> -In Definition 2.2, clarify that the random subset R(x) is drawn so that the marginal probability of i \in R(x) = x_i.
>
> We have made this explicit, thank you for the catch.
>
> -In line 246, typo - 1 - x_j(1-x_j / 2)
> -In line 261, “no greedy OCRS cannot” -> “no greedy OCRS can”
>
> Thank you for catching these typos, we have fixed them.

---

### Official Review · Reviewer_LXMv · 2022-07-12

**Rating:** 6
**Confidence:** 3
**Soundness:** 3 good
**Presentation:** 2 fair
**Contribution:** 3 good

**Summary:**

The paper presents some results regarding online contention resolution schemes. The main result, to my taste, is the result for selecting a single item. One can rephrase this result as follows. We are given a point x in the simplex (\sum x_i \le 1) and then draw a random set of “elements” by taking each one with probability x_i independently. The elements will then be presented to us in arbitrary order and we have to select one upon seeing it. The question is: What is the best number c such that the probability of selecting any element is at least cx_i? The paper proves that the best such c is 1/e. Nice! The way to do it is as follows. You draw this random set, then clean it up by dropping each element on it with probability 1-x_i/2, and finally take the first element that was not dropped. This can be implemented as a greedy OCRS and therefore it improves upon the factor 1/4 of FSZ16. Also, it is not too hard to show this is best possible. One should note that this clean-up step is needed as otherwise we may have a “large” probability element that may come first and then would block the remaining “small” probability elements from ever appearing. Say x_1=1-eps and x_2=eps, if we do not clean-up we will only get element 2 w.p. eps^2.

The paper also extends this idea to some classes of transversal matroids.

**Questions:**

No further questions

**Ethics Review Area:**

["I don’t know"]

**Strengths And Weaknesses:**

The main result is nice and simple. I appreciate that. The authors also show it can be useful in related more complex settings though that part did not feel like very exciting to me.

I do not like the write-up very much. It is written for specialists and not for a wide audience. Even the problem definition is not clearly stated and there are missing definitions (e.g. that of R(x)).

I think the authors should try to rephrase their main result in simple terms (in particular without using the OCRS terminology) in the first few pages of the paper. This is relatively easy. And then, once they really need matroids and more general CRS they can give more notation and technicalities.

---

> ### Author Response · Authors · 2022-08-02
> **Rebuttal Response**
>
> We would like to thank the reviewer for taking the time to read our paper and thank them for their feedback.
>
> As per the reviewer's suggestions, we have revised the write-up and we have made an attempt to present the main result in the first few pages of the introduction in a simpler way. We hope this change makes the paper easier to read and thus more appealing to a wider audience. We have also added clarifications for the missing definitions.

---

### Official Review · Reviewer_Wsp9 · 2022-07-15

**Rating:** 4
**Confidence:** 3
**Soundness:** 4 excellent
**Presentation:** 3 good
**Contribution:** 2 fair

**Summary:**

The paper considers greedy online contention resolution schemes. A contention resolution scheme (CRS) is, roughly, a method to round a fractional solution for a (packing) problem to a random integral one. More precisely, given a vector $x\in[0,1]^n$ that is feasible for a fractional relaxation of a problem, let $y\in\{0,1\}^n$ be an integral vector obtained by selecting each $y_i$ to be $1$ independently with probability $x_i$. But $y$ may be infeasible for the problem, so now the task of the CRS is to switch some of the $y_i$ from 1 to 0 in order to obtain a feasible solution (in other words, the CRS selects an integral $z\le y$ that is feasible). A CRS is *online* (abbrev. OCRS) if the $y_i$ are revealed one by one, and the OCRS has to pick $z_i$ immediately after $y_i$ is revealed. OCRSs have applications to various online problems such as prophet inequalities, stochastic probing etc. An OCRS is said to be $c$-selectable if, roughly, $P(z_i=1)\ge c\cdot x_i$ for each $i$ (the precise definition given in the paper is somewhat more complicated, which I believe has to do with the type of ``almighty adversary’’ considered, which can choose the order in which the entries of $y_i$ are revealed with full knowledge of all random number generators). Intuitively, a $c$-selectable OCRS corresponds to losing a factor $c$ in the rounding procedure.

The paper considers a subtype of OCRSs that are called *greedy*, which means that it first chooses (randomly) a downwards-closed family of coordinates, corresponding to a subset of feasible solutions, and then greedily tries to set each $z_i$ to 1 if possible while maintaining a solution corresponding to this downwards-closed family.

Results:
- They give a 1/e-selectable greedy OCRS for the single-item setting (i.e, the feasibility constraint is $\sum_i x_i\le 1$). This improves over the previous 1/4-selectable greedy OCRS, but is worse than the known 1/2-selectability achieved by a non-greedy OCRS. It also implies immediately the same result for partition matroids.
- They show that 1/e is tight for greedy OCRSs for the single-item setting.
- They extend their result to so-called transversal matroids.

In an experimental evaluation, their 1/e-selectable greedy OCRS indeed improves upon the previous 1/4-selectable greedy OCRS.

**Questions:**

A discussion about benefits of greedy over non-greedy OCRSs would be useful. Are they significantly simpler? Does the 1/2-selectability of the known non-greedy OCRS also hold against almighty adversaries?

Specific (minor) comments:
- Line 172: Is $\mathcal P_{\mathcal I}$ the same thing that you denoted as $\mathcal P$ before?
- Line 175: What is $R(x)$? I later found kind of a definition of $R$ in the introduction, but it would have been good to define it in preliminaries.
- Line 211, 237 and elsewhere: You omit some of the subscripts of $\mathcal F$. It would have been good to state explicitly that you may drop them from the notation.
- Line 234: Here you seemingly use $R$ with a different meaning than before.
- Line 246: One of the $x_i$ should be $x_j$.
- Line 274: Presumably $R_v$ only contains neighbors of $v$? Otherwise, how would you conclude that each $S\in\mathcal F$ is covered by a matching?
- Line 302 you say that number of iterations is set to 200,000, but in the caption of Figure 1 it states 10,000 instead of 200,000.
- Figure 2: What is the number on the x-axis? I thought it’s the size of $N(u)$, but it goes up to 4 whereas in the description of the setup you say it goes only up to 3.

**Limitations:**

This is a theoretical work and negative societal impact is not expected. Limitations are clear from the mathematical formulations.

**Strengths And Weaknesses:**

The main result — improving from 1/4 to 1/e and showing tightness — is the final answer for single-item greedy OCRSs. This is a nice and clean result. I’m not entirely sure about the significance though, given that a (better) 1/2-selectable non-greedy OCRS is known. Why would one choose this greedy OCRS when the known non-greedy one performs better? The experimental evaluation only contains comparison with the previous (suboptimal) greedy one, but not with the better non-greedy one.

The writing is mostly good, but some things were unclear to me for some time because of insufficient definitions (see below; but I could eventually guess the likely intended meaning).

---

> ### Author Response · Authors · 2022-08-02
> **Rebuttal Response**
>
> The reviewer raises several important questions regarding the benefits of using a greedy vs a non-greedy OCRS and we would like to thank them for the valuable feedback and the opportunity to provide some more detail about this. As the reviewer observed, greedy OCRSs are inherently much simpler than non-greedy OCRSs. Usually, to obtain a non-greedy OCRS for a non-trivial constraint, one has to go through the dual of an LP which captures the problem, as is the approach of Lee and Singla [LS18]. This leads to a non-intuitive algorithm which, in many situations, can be difficult to implement (i.e. when the OCRS is used for a problem from Algorithmic Game Theory, in which one has to account for the agents' strategic behaviour).
> Greedy OCRSs are simpler and more intuitive.
>
> Furthermore, the known $1/2$-selectable OCRS due to Lee and Singla [LS18] does not provide any guarantees against an almighty adversary who has knowledge of the future realizations and the algorithm's randomness, but rather against an online adversary who at any point in time has the same information as the algorithm. In this context, for the constraints considered, our OCRS provides a stronger guarantee, since it holds against a more powerful adversary.
>
> In addition, we would like to point out an application which crucially requires the OCRSs used to be greedy. Between the submission time and now, we were made aware of a paper titled "Delegated Pandora's Box" by Bechtel, Dughmi and Patel. They study the "delegation gap" of the generalized Pandora's box problem and in fact reduce the problem to the design of an OCRS which is necessarily greedy. In their model, they need a greedy OCRS since the reduction forces the algorithm to play against the almighty adversary. This result is the first example of an application in which non-greedy OCRSs cannot be applied and a greedy OCRS is needed. We have revised the write-up and now cite this paper.
>
> Addressing the minor comments:
> -Line 172: Is $\mathcal{P}_\mathcal{I}$ the same thing that you denoted as $\mathcal{P}$ before?
>
> Yes, we should have made it explicit; $\mathcal{P}$ is defined as a polyhedral relaxation of a constraint $\mathcal{C}$ and $\mathcal{I}$ is the set of independent sets of $\mathcal{C}$, thus $\mathcal{P}$ depends on $\mathcal{I}$.
>
> -Line 175: What is $R(x)$? I later found kind of a definition of $R$ in the introduction, but it would have been good to define it in preliminaries.
>
> $x$ is a vector in $[0,1]^n$, and $R(x)$ is a random set from $2^\mathcal{N}$ in which each element $i$ appears with probability $x_i$. We have added this clarification.
>
> -Lines 211, 237 and elsewhere: You omit some of the subscripts of $\mathcal{F}$. It would have been good to state explicitly that you may drop them from the notation.
>
> Thank you for the suggestion, we have added a note that the subscript of $\mathcal{F}$ is dropped when it is clear from context.
>
> -Line 234: Here you seemingly use $R$ with a different meaning than before.
>
> It is actually the same meaning, but we agree it would be more illuminating if we wrote $R(q)$ instead of $R$ to make it explicit. We have implemented this change.
>
> -Line 246: One of the $x_i$ should be $x_j$.
>
> Thank you for the catch, we have fixed it.
>
> -Line 274: Presumably $R_v$ only contains neighbors of $v$? Otherwise, how would you conclude that each $S \in \mathcal{F}$ is covered by a matching?
>
> Correct, $R_v$ only contains neighbors of $v$, to satisfy the given feasibility constraints. We have made sure this is explicit.
>
> -Line 302: you say that number of iterations is set to 200,000, but in the caption of Figure 1 it states 10,000 instead of 200,000.
>
> We had forgotten to update the caption of the figure from an earlier run of 10,000 iterations. Thank you for catching this.
>
> -Figure 2: What is the number on the x-axis? I thought it’s the size of $N(u)$, but it goes up to 4 whereas in the description of the setup you say it goes only up to 3.
>
> You are correct, the x-axis corresponds to $|N(u)|$, the size of the neighborhoods. We had initially performed experiments for $|N(u)| = 2$ and $3$, but decided to add another value of $|N(u)| = 4$ to provide more clarity. Unfortunately, we had not updated the text accordingly, but we have updated it now.

---

### Author Response · Authors · 2022-08-02
**Rebuttal General Comment**

We would like to thank all reviewers for carefully reading our paper, and for providing valuable comments and suggestions. We have implemented their suggestions, fixed any typos or missing definitions and have revised the paper. We address the main questions raised by each reviewer separately.

We would like to once again sincerely thank all the reviewers for their time spent reading our paper and their valuable and helpful input.

---

### Meta-Review · Area_Chair_nbe3 · 2022-08-25

**Recommendation:** Accept
**Confidence:** Certain

**Metareview:**

Executive summary:

The paper considers the design of greedy online contention resolution schemes (OCRS) for the single-item setting and certain matroids (partition matroids, transversal matroids). The main result is that there is a 1/e-selectable greedy OCRS (which improves over the best known bound of 1/4 for greedy OCRS), and that this is best possible.

Discussion and recommendation:

This is a nice little result. Not tremendously difficult, but fundamental. A plus is that the question is resolved tightly. All but one reviewer felt positively about the paper. A major concern raised in the reviews was that it's unclear why we care about greedy OCRS. In the rebuttal, the authors emphasized that greedy OCRS yield guarantees against an almighty adversary and that they recently found application in a delegation variant of the Pandora's Box problem (Bechtel, Dughmi, and Patel [EC'22]).

(Weak) accept.

---

Additional comments:

I always thought of OCRS as one of the two main techniques that have emerged for proving prophet inequalities and guarantees for posted-price mechanisms; the other being the "balanced prices" framework. I would encourage to extend the discussion in the related work accordingly, and cite the most relevant works on the "balanced prices" framework. Or, at least, cite the most relevant papers in that direction (see list below).

Citations to add:

Kleinberg and Weinberg. Matroid Prophet Inequalities. STOC'12.

Feldman, Gravin, Lucier. Combinatorial Auctions via Posted Prices. SODA'15.

D\"utting, Feldman, Kesselheim, Lucier. Prophet Inequalities made Easy: Stochastic Optimization by Pricing Non-Stochastic Inputs. FOCS'17.

D\"utting, Kesselheim, Lucier. An O(log log m) Prophet Inequality for Subadditive Combinatorial Auctions. FOCS'20.

**Award:**

No

---

### Decision · Program_Chairs · 2022-09-14

Accept